# Thermal Requirements of *Ooencyrtus submetallicus* (Hym.: Encyrtidae) and *Telenomus podisi* (Hym.: Platygastridae) Parasitizing *Euschistus heros* Eggs (Hem.: Pentatomidae)

**DOI:** 10.3390/insects12100924

**Published:** 2021-10-09

**Authors:** Valeria Freitas Chaves, Fabricio Fagundes Pereira, Jorge Braz Torres, Ivana Fernandes da Silva, Patrik Luiz Pastori, Harley Nonato de Oliveira, Valmir Antônio Costa, Carlos Reinier Garcia Cardoso

**Affiliations:** 1Faculdade de Ciências Agrárias, Universidade Federal da Grande Dourados, Rodovia Dourados/Itahum, Km 12, Dourados 79804-970, MS, Brazil; valeriafreittas1@gmail.com; 2Faculdade de Ciências Biológicas e Ambientais, Universidade Federal da Grande Dourados, Rodovia Dourados/Itahum, Km 12, Dourados 79804-970, MS, Brazil; ivanaf.silva@hotmail.com (I.F.d.S.); cr.garcia.cardoso@gmail.com (C.R.G.C.); 3Departamento de Agronomia, Universidade Federal Rural de Pernambuco, R. Dom Manoel de Medeiros, s/n, Recife 52171-900, PE, Brazil; jorge.torres@ufrpe.br; 4Departamento de Fitotecnia, Universidade Federal do Ceará, Av. Mister Hull, Fortaleza 60356-001, CE, Brazil; plpastori@ufc.br; 5Empresa Brasileira de Pesquisa Agropecuária, Embrapa-CPAO, Rodovia BR 163, Dourados 79804-970, MS, Brazil; harley.oliveira@embrapa.br; 6Instituto Biológico, Agência Paulista de Tecnologia dos Agronegócios, Secretaria de Agricultura e Abastecimento, Alameda dos Vidoeiros, 1097-Sítios de Recreio Gramado, Campinas 13101-680, SP, Brazil; valmircosta@gmail.com

**Keywords:** biological control, egg parasitoid, biological characteristics, brown stink bug

## Abstract

**Simple Summary:**

The brown stink bug, *Euschistus heros,* is the most abundant species infesting Brazilian soybean crops, causing significant yield losses. This stink bug is controlled with chemical insecticides, although the use of biological control with the egg parasitoids *Ooencyrtus submetallicus* and *Telenomus podisi* is increasing. Our objective in this study was to evaluate the development of *O. submetallicus* and *T. podisi* in *E. heros* eggs at different temperatures, and to estimate the number of annual generations for seven representative soybean-producing regions in Brazil. In the comparative biology study, the sex ratio and individuals per egg were higher for *O. submetallicus* in all regions. In the study of thermal requirements, the temperatures of 16, 19, 22, 25, 28, 31, and 33 °C were tested. The base temperature (Tb) for *O. submetallicus* and *T. podisi* was found to be 9.3 and 6.7 °C, and the thermal constant (K) was 336.9 and 272.7 degree-days, respectively. The average number of annual generations was estimated from the average temperatures of the last ten years in each region; based on these results, both parasitoids presented a higher number of generations than the host in all regions. *O*. *submetallicus* and *T. podisi* have the same ability to parasitize and develop in the eggs of *E. heros* under the conditions studied; in addition, both exhibited development and satisfactory parasitism at temperatures between 19 and 31 °C. The development of the two parasitoids was faster than *E. heros,* and the number of generations was higher for the seven studied localities.

**Abstract:**

Temperature is one of the abiotic factors that strongly influences the biology and behavior of insects. In this study, we assessed the development of egg parasitoids *Ooencyrtus submetallicus* and *Telenomus podisi* parasitizing *Euschistus heros* eggs at different temperatures, and estimated the average number of generations for seven representative soybean-producing regions in Brazil. A comparative biology study was conducted, where the percentage of parasitism and emergence, life cycle duration (egg–adult), and longevity were found to be similar between *O. submetallicus* and *T. podisi*. The sex ratio and the number of individuals emerged from parasitized egg were higher in *O. submetallicus* in all regions. In the study of thermal requirements, temperatures of 16, 19, 22, 25, 28, 31, and 33 °C were tested. *O. submetallicus* and *T. podisi* developed at temperatures between 16 and 31 °C. The base temperature (Tb) for *O. submetallicus* and *T. podisi* was 9.3 and 6.7 °C, respectively; the thermal constant (K) was 336.9 and 272.7 degree-days, respectively. The estimated average number of annual generations of both parasitoids was higher than the host. *O. submetallicus* and *T. podisi* have the same ability to parasitize and develop in eggs of *E. heros* under the conditions studied. Both parasitoids exhibited satisfactory development and parasitism at temperatures between 19 and 31 °C. The development of both parasitoids was faster than their host, and the number of generations was higher for the seven studied localities.

## 1. Introduction

The tropical climate in Brazil favors the cultivation of soybeans, *Glycine max* (L.) Merrill, in different locations, as well as the incidence of herbivorous species reaching the status of pest [1]. The brown stink bug, *Euschistus heros* (Fabricius, 1794) (Hemiptera: Pentatomidae), stands out amongst the pest herbivore species. This species is the most abundant Pentatomid (hereafter, stink bug) pest species of soybean, occurring from northern to southern Brazil [2]. Nymphs and adults feed on grains in development, causing drastic reductions in grain quality and yield. During feeding, the stink bugs deliver toxins into the plant, causing physiological disturbances such as leaf retention, which hinders harvesting [3]. 

Stink bugs are almost exclusively controlled with synthetic insecticides, both via seed treatment and spraying on plants, with implications for the environment and human health [4]. Therefore, more appropriate methods based on the principles of integrated pest management (IPM) have been pursued. Biological control has become an important method to reduce the population of agricultural pests, including the use of egg parasitoids [5]. The egg parasitoid *Ooencyrtus submetallicus* (Howard, 1897) (Hymenoptera: Encyrtidae) is reported in the literature as an egg parasitoid of stink bugs, such as *Nezara viridula* (Linnaeus, 1758) [6], *Edessa meditabunda* (Fabricius, 1974) [7], and *E. heros* [8]. The parasitoid *Telenomus podisi* (Ashmead, 1893) (Hymenoptera: Platygastridae) was shown to be the most effective at controlling *E. heros* [9].

The efficiency, development, and reproduction of these parasitoids can be affected by abiotic factors such as light, humidity, and especially temperature, which is the factor that most influences biological aspects [10,11]. The body temperature of insects exposed to very high temperatures increases to lethal levels, whereas lower temperatures can cause physical damage, negatively affecting insect development [12], or compromise rates of development, emergence, parasitism, fecundity, and survival [13]. Thus, having knowledge of the ideal insect development temperature is essential for developing programs for the mass creation of parasitoids [14]; from information regarding the ability to adapt, it is possible to establish the optimal temperature for parasitoid development, estimate the number of generations for the regions in which they are located, and predict their establishment in the field [15]. 

The effects of temperature variation on natural enemies can be studied by determining thermal requirements, where the needs of each parasitoid in their hosts are considered, such as the lower thermal threshold (base temperature), the one below which no development occurs; the upper thermal threshold, being the maximum growth temperature; the optimum temperature, usually corresponding to the fastest development; and greater number of descendants. From the thermal constant (degree-days), which is the required number of accumulated degree-days for the insect to transition from one stage to another, and the temperature of the site under study, it is possible to estimate the number of generations that can be produced under certain environmental conditions [16,17]

Several studies have been conducted with the parasitoid *T. podisi* to determine the ideal temperature for its creation and release in the field, including estimating the number of generations of this species [17,18,19,20], since this species is found in several agroecosystems, parasitizing the eggs of various Pentatomids, showing adaptation to different climatic conditions and hosts [21]. For the parasitoid *O. submetallicus,* no studies have been conducted yet regarding the ideal temperature for its creation and release in the field.

Therefore, the development of *O. submetallicus* and *T. podisi* parasitizing *E. heros* eggs was assessed and the thermal requirements of these parasitoids were determined. The outcome allowed the estimation of the average number of generations per year for seven representative soy-producing locations in central-western and southern Brazil.

## 2. Materials and Methods

The experiments were conducted on the premises of the Laboratory of Biological Control of Insects (LECOBIOL) (22°19’80″ S,54°93’38″ W), the Faculty of Biological and Environmental Sciences (FCBA), the Federal University of Grande Dourados (UFGD), in Dourados, Mato Grosso do Sul, Brazil.

### 2.1. Rearing and Maintenance of Insects Used in the Experiments

#### 2.1.1. *Ooencyrtus submetallicus*

*Ooencyrtus submetallicus* was reared following the methodology described in Faca [22]. Briefly, the parasitoids were kept in glass tubes (2.0 × 15.0 cm (diameter × height)) closed with cotton, containing a droplet of bee honey for feeding. The parasitoids were reared in *E. heros* eggs until the experiments were assembled. One colony was maintained in an air-conditioned B.O.D.-type chamber (model EL 222, ELETROLab^®^, São Paulo, SP, Brazil) and the other in an air-conditioned room; both colonies were maintained at a temperature of 25 ± 2 °C, 70% ± 10% relative humidity, and 14:10 h (light:dark) photoperiod. The species of this parasitoid was found in *E. meditabunda* eggs in the leaves of tomato, *Solanum lycopersicum* (Linnaeus, 1753) (Solanales: Solanaceae). The collection was performed by Antonio de Souza Silva in the region of Dourados, MS, Brazil.

#### 2.1.2. *Telenomus podisi*

Adults of *T. podisi* were supplied by Embrapa Agropecuária Oeste, from Koppert Biological Systems. The parasitoids were kept in glass tubes (2.0 × 15.0 cm (diameter × height)) closed with cotton, containing a droplet of bee honey for feeding. The parasitoids were reared in *E. heros* eggs until the experiments were assembled. Part of the colony was kept in an air-conditioned room and the other in an air-conditioned B.O.D.-type chamber (model EL 222, ELETROLab^®^, São Paulo, SP, Brazil); both colonies were maintained at a temperature of 25 ± 2 °C, 70% ± 10% relative humidity, and 14:10 h (light:dark) photoperiod.

#### 2.1.3. *Euschistus heros*

The brown stink bug was collected from the experimental farm of UFGD through a scanning network and manual collection. Nymphs and adults were reared in cages made of 5 L transparent plastic pots. For feeding, we provided 500 g of fresh pods of beans *(Phaseolus vulgaris* L.), 500 g of green seeds of ligustro (Ligustrum sp.), 50 g of dried soybean grains *(Glycine max* L.), 50 g of raw and peeled peanuts (*Arachis hypogaea* L.) with water provided by a cotton swab moistened with distilled water, placed in plastic Petri dishes perforated in the middle. Food was exchanged every five days. A 30 cm diameter filter paper folded into a fan shape was used as the oviposition substrate for the stink bugs [23]. Part of the eggs that were collected daily were separated to maintain the stock creation of *E. heros;* these were placed in Petri dishes with a wet cotton wad and bean pod, another part was assigned to the multiplication of parasitoids *O. submetallicus* and *T. podisi* and the experiment. The rearing was conducted in an acclimatized room at 25 ± 2 °C, 70% ± 10% RH, and 14:10 (light:dark) photoperiod. *E. heros* was identified by Dr. Jocélia Grazia, taxonomist of Pentatomid species (Federal University of Rio Grande do Sul—UFRGS).

### 2.2. Experimental Development and Analysis

#### 2.2.1. Comparative Biology of *O. submetallicus* and *T. podisi*

We fixed 24 h old *E. heros* eggs on sky blue cardboard (0.50 × 1.00 cm (width × length)) with Arabic gum diluted in water (20%) at the density of 10 eggs per card. Each card was inserted into glass tubes (1.0 × 9.50 cm (diameter × height)) containing a drop of pure bee honey on the inner wall of the tube as food for the parasitoid. The parasitism was achieved by releasing one 120 h old female *O. submetallicus* [22] inside the tube, which was closed with cotton. The same procedure was performed for the females of *T. podisi,* however, at 24 h of age. The parasitism of eggs by the females was allowed for 24 h. After this period, the females of both parasitoids were removed from the tubes, and the cards containing eggs were transferred to the climate chamber set at 25 ± 2 °C, 70% ± 10% RH, and 14:10 h (light:dark) photoperiod until the possible emergence of the adult *O. submetallicus* and *T. podisi*, thus confirming the parasitism. 

The characteristics evaluated during this study were developmental time (egg–adult), percentage of parasitism ((number of dark eggs/total number of eggs offered) × 100), emergence percentage ((number of eggs with emergence hole/number of dark eggs) × 100), number of individuals emerged per egg parasitized, sex ratio (sr = number of females/total number of offspring), and longevity of offspring with and without food (bee honey). The experimental design was completely randomized (DIC), with two treatments (parasitoid species) and 20 replications (one female parasitizing 10 eggs each). Data were subjected to analysis of variance and Student’s *t*-test at 5% probability of error.

#### 2.2.2. Effect of Temperature on the Development of *O. submetallicus* and *T. podisi*

*E. heros* eggs less than 24 h old were individualized in glass tubes (1.0 × 9.50 cm (diameter × height) containing a droplet of bee honey deposited on the inner wall of the tube and closed with a cotton swab. Each tube contained a card with 10 eggs received two 120 h old female *O. submetallicus*. The same procedure was performed for *T. podisi*, but with 24 h old females. Parasitism was allowed for 5 h in an acclimatized room at 25 ± 2 °C and 70% ± 10% RH. At the end of this period, the females were discarded, and the tubes holding the cards with eggs were transferred to climatized chambers regulated to 16, 19, 22, 25, 28, 31, or 33 °C, and a 14:10 h (light:dark) photoperiod.

The experimental design was completely randomized, with seven treatments (temperatures) with 20 replications each (10 eggs each). For each temperature, the following characteristics were determined: developmental time (egg–adult in days), determined through daily observations, always at the same time; percent of emergence ((no. of eggs with emergence hole/no. of dark eggs) × 100); number of individuals emerged per egg parasitized; and sex ratio (sr = no. of females/no. total). The data obtained were subjected to analysis of variance, and when significant at 5% probability, regression analysis was performed. The equation that best fit the data was chosen based on the coefficient of determination (R^2^) and the significance of the regression coefficients (βi).

#### 2.2.3. Estimates of the Thermal Requirements and Number of Generations of *O. submetallicus* and *T. podisi*

The lowest temperature threshold (Tb) and the thermal constant (K) were estimated by the hyperbola method [24] based on the developmental time (egg–adult) of *O. submetallicus* and *T. podisi* at the evaluated temperatures. The number of annual generations of the two parasitoid species was estimated for the representative soybean-producing regions in central-western and southern Brazil, including Tupanciretã, Rio Grande do Sul, RS; Marshal Cândido Rondon, Paraná, PR; Maracaju, Mato Grosso do Sul, MS; Dourados, Mato Grosso do Sul, MS; Ponta Porã, Mato Grosso do Sul, MS; Sorriso, Mato Grosso, MT; and Rio Verde, Goiás, GO. The equation: NG = (T(Tm-Tb)/K) was employed, where K is the thermal constant, Tm is the average temperature for each region, Tb is the lowest temperature threshold, and T is the time considered in days. Climatic data for the last ten years of the regions were provided by the National Institute of Meteorology (INMET).

## 3. Results

### 3.1. Comparative Biology of O. submetallicus and T. podisi

The percentages of parasitism and emergence were statistically similar among the parasitoids: from 59.0% to 62.5% and from 86.4% to 87.3% for *O. submetallicus* and *T. podisi*, respectively. We also found no difference in the life cycle duration (egg–adult) between the parasitoids (Table 1). The proportion of females in the offspring, however, was significantly different with 100% females in the offspring of *O. submetallicus* and 54% females of *T. podisi*. The number of parasitoids that emerged per parasitized egg was also superior for *O. submetallicus* (Table 1). The adult longevity was similar for the produced parasitoids regardless of being fed pure bee honey or not. The adults of both species maintained on food lived, on average, 50% longer (Table 1).

### 3.2. Effect of Temperature on the Development of O. submetallicus and T. podisi

The developmental time (egg–adult) of the parasitoids presented an inverse relationship with increasing temperature. For *O. submetallicus,* the developmental time was 33.4 ± 0.25 days at 16 °C and 13.2 ± 0.09 days at 33 °C. The same pattern was observed for *T. podisi*, which was 35.4 ± 0.11 and 11.3 ± 0.11 days, when reared at 16 and 33 °C, respectively (Figure 1). The emergence of *O. submetallicus* was 74.2% and 88.8% at 28 and 33 °C, respectively; *T. podisi* achieved 92.9% emergence at 25 °C and 72.5% at 31 °C (Figure 2).

The number of offspring produced per host egg parasitized by *O. submetallicus* was not influenced by the temperature regimes, with an average of 1.31 parasitoids/egg. However, the temperature affected the number of individual *T. podisi* per host egg parasitized (Figure 3). The proportion of females in the offspring of *O. submetallicus* was 100% for all temperatures tested, whereas for *T. podisi*, the proportion of females in the offspring ranged from 54% to 67%, which were observed at temperatures 25 and 33 °C, respectively (Figure 4).

### 3.3. Thermal Requirements and Estimate of the Number of Generations of O. submetallicus and T. podisi

The thermal requirements of *O. submetallicus* parasitizing *E. heros* eggs were determined based on the model y = (1/D) = −0.0020098 + 0.002968x (r^2^ = 0.98), with values for the lowest temperature threshold (Tb) estimated as 6.77 °C and thermal constant (K) estimated as 336.97 degree-days (DD) (Figure 5). For *T. podisi*, y = (1/D) = −0.034174 + 0.003670x (r^2^ = 0.99), the lowest temperature threshold (Tb) was 9.31 °C and the thermal constant (K) was determined as 272.46 degree-days (DD) (Figure 6).

The estimated annual generations of *O. submetallicus* for Dourados, MS; Ponta Porã, MS; Maracaju, MS; Sorriso, MT; Rio Verde, GO; Marechal Cândido Rondon, PR; and Tupanciretã, RS were 18.33, 16.39, 15.26, 18.78, 14.92, 13.36, and 11.99, respectively; for *T. podisi,* the estimations were 18.29, 15.29, 15.47, 18.82, 15.05, 13.12, and 11.43 generations per year, respectively. For the host pest species *E. heros* for these same locations, the outcomes were 9.76, 8.57, 7.41, 11.03, 7.06, 4.05, and 5.46 generations per year, respectively (Table 2).

## 4. Discussion

The results highlight the potential of the egg parasitoid *O. submetallicus* for use in the biological control of the brown stink bug, *E. heros*. The percentage of parasitism and other biological characteristics (e.g., produced offspring and percentage of females in the offspring) of *O. submetallicus* were similar or superior to the commercial parasitoid *T. podisi* in parasitizing *E. heros* eggs. 

The emergence percentages observed for both species were considered adequate, because above 72% is ideal for laboratory rearing [25]. A high emergence percentage of parasitized eggs indicates that the host is satisfactory for the parasitoids to complete their development, which is an important outcome for biological control programs [26]. Thus, *E. heros* eggs are suitable for rearing these parasitoids in the laboratory, and future semi-field and field studies are needed to confirm their ability to search for and parasitize the host.

The developmental time of both parasitoid species was similar when reared in *E. heros* eggs (average of 17 days), requiring only 60% of the time to complete the development of their host, which corresponds to 28.4 days at 25 °C. This result is an interesting feature for both parasitoids and a favorable factor with respect to field releases [27]. The genus *Ooencyrtus* is known to deposit more than one egg per host and is not selective regarding when it will parasitize or even super-parasitize its hosts. Furthermore, this parasitoid is smaller than other parasitoids, including *T. podisi* [28].

The longevity of parasitoids consists of their survival period and emergence death; when this characteristic is determined, considering field stress conditions such as temperature and food scarcity can provide information on the range of releases for regulating the target pest [25]. The parasitoids *O. submetallicus* and *T. podisi* survived longer when fed. Variations in parasitoid longevity may be related to several factors, ranging from host eggs, environmental conditions to which they are exposed, and energy expenditure during copulation, to oviposition and food deprivation [29].

The offspring of *O. submetallicus* under the study conditions were 100% female, which is an advantage for the population growth of the species and for biological control programs, because females are directly responsible for parasitism and, consequently, production of new offspring and reduction of the pest population [12]. The sex ratio of *O. submetallicus* can be explained due to its type of reproduction, which occurs by thelium thendogenesis, where the eggs develop only female offspring [6]. In biofactories, the predominance of females over male parasitoids is important for large-scale breeding, since they are mainly responsible for the following generations in the laboratory [30].

Temperature influenced the characteristics evaluated for *O. submetallicus* and *T. podisi.* Temperatures of 16 and 33 °C had a stronger impact on the duration of the egg–adult cycle, emergence, number of individuals per egg, and sex ratio. Notably, even though temperature is considered extremely important for the success in biological control programs, other abiotic and biotic factors may be responsible for changes in these characteristics, such as relative humidity, photoperiod, and interspecific and intraspecific competition [5,31,32].

According to the results, the inverse relationship between temperature and the duration of the egg–adult cycle can be explained by the low metabolic activity of the parasitoids when they are at low temperatures; thus, the development is completed at lower speed. Conversely, with the increase in temperature, metabolic activity increases and development is completed with faster speed, which can be an advantage for biological control since adults will appear earlier; however, this can have negative effects on the parasitism of natural enemies [19]. At very high temperatures, the ideal time of development of these biological agents is impaired and mortality increases [12]. Similar inverse relationships were observed for *T. podisi* parasitizing eggs *of Podisus nigrispinus* (Dallas) (Hemiptera: Pentatomidae) [21] and *Telenomus pachycoris* (Johnson, 1984) (Hymenoptera: Platygastridae) parasitizing eggs of *Pachicoris torridus* (Scopoli, 1772) (Hemiptera: Scutelleridae) [17].

In the temperature range between 22 and 28 °C, the highest number of adults emerged, which may indicate the adaptability of the parasitoids. The lower percentage of emergence of *O. submetallicus* and *T. podisi* at 33 °C supports the hypothesis that the exposure of parasitoids to higher temperatures throughout their development negatively affects their life history, evidencing that development tends to be slower when occurring outside the optimal temperature range [33,34], reducing the emergence at temperatures below 16 °C. Similar results were reported for *T. podisi* [21,35] and *Ooencyrtus mirus* (Triapitsyn, 2021) (Hymenoptera: Encyrtidae) when they were submitted to a temperature of 15 °C [36]. However, very low or high temperatures in general do not occur for long periods in the field; therefore, field tests still need to be performed to further understand the real effect of weather conditions on *O. submetallicus* and *T. podisi.*

For *O. submetallicus,* there was no influence of temperature on the sex ratio, with 100% female offspring. According to Wilson and Woolcock [6], under temperature conditions higher than 29 °C, it is possible produce male offspring, a finding not observed in this study. The sex ratio for *T. podisi* was influenced by temperature, mainly at 16 and 33 °C, where there was a higher proportion of females than males. Similar results were obtained for *T. podisi* at temperatures below 15 °C, where only female emergence occurred [37]. This characteristic is considered satisfactory for both parasitoids at all studied temperatures because it presents a higher number of females in relation to males, an important factor in mass rearing systems and in the selection of individuals for field release as females are responsible for parasitism and the next generation [26].

Based on the duration of the cycle development (egg–adult) at the different temperatures tested, the thermal constant (K) and the lower thermal limit of development (Tb) were determined. For *O. submetallicus,* the thermal constant was higher, indicating that it requires greater thermal accumulation (degree-days) to complete its development, different from *T. podisi,* which needs less heat accumulation. However, *O. submetallicus* has superior daily thermal accumulation potential, since *T. podisi* presented a higher base temperature. This means that there are no differences in the duration of the egg–adult cycle between the two species. The thermal requirements in the present study are similar to those found for *T. podisi* in *Podisus nigrispinus* eggs (Dallas, 1851) (Hemiptera: Pentatomidae), with a base temperature of 11.1 °C and thermal constant of 205.3 DD [21]. However, the thermal requirements were lower for *Telenomus remus* (Nixon, 1937) (Hymenoptera: Platygastridae) in the eggs of *Spodoptera frugiperda* (Smith, 1797) (Lepidoptera: Noctuidae), where the base temperature was 12.5 °C and the thermal constant was 158.9 DD [12]. This variation between the values found, both for base temperature and thermal constant for the two species studied, can better characterize its effects, if considered in development estimates, that is, the number of generations, for a long period when temperature becomes a limiting factor [21]. 

*Ooencyrtus submetallicus* and *T. podisi* develop faster than their host *E. heros* at the studied temperatures, since it presented a thermal requirement higher than those of the parasitoids, being a Tb of 14.2 °C and K of 327.8 DD [18]. The number of generations of both parasitoids was also higher than that of the host under the temperature conditions studied for the selected soybean-producing localities of the midwest and south of Brazil (Ponta Porã-MS, Dourados-MS, Maracaju-MS, Tupanciretã-RS, Mal. Cândido Rondon-PR, Sorriso-MT, and Rio Verde-GO). The findings that development is faster and the number of generations of parasitoids is higher than those of their host at the same temperature are important characteristics for biological control programs, since this can help with the stabilization and/or reduction of the pest population in the field [30]. However, other factors can influence the number of generations, such as photoperiod, air humidity, quality, and host availability; when these conditions are favorable, temperature can become a limiting factor for insects [36,38]. The results found for *O. submetallicus* compared to *T. podisi* in *E. heros* eggs are important, demonstrating that, under controlled laboratory conditions, it is possible to manipulate/control temperature to decrease or increase the parasitoid development cycle in order to synchronize with emergence. However, it is important to emphasize that the results were obtained in a laboratory under constant temperature; therefore, studies with fluctuating temperatures are necessary to understand the real effect of temperature on the characteristics of parasitoids, to mimic the daily fluctuations in temperature that occur in the field. In studies conducted for *Diaphorencyrus aligarhensis* (Shafee, Alam and Argarwall, 1975) (Hymenoptera: Encyrtidae) and *Tamarixia radiata* (Waterston, 1922) (Hymenoptera: Eulophidae), fluctuating temperatures demonstrated significant effects on the life history of parasitoids, helping in the optimization of mass creation and release, in addition to improving climate modeling predictions [39,40].

## 5. Conclusions

In summary, from the results of present study, it is possible to better understand the effect of an abiotic factor, temperature, on the biological characteristics of *O.*
*submetallicus* and *T.*
*podisi* in *E. heros* eggs; we also demonstrated that both have a similar ability to reproduce in *E. heros* eggs. The results obtained demonstrate the ability of the parasitoids to develop in temperature ranges similar to those studied. We also obtained important information about the multiplication of these parasitoids under laboratory conditions, and even large-scale production could be achieved based on knowledge of their interaction with the host, their doubling time to the next generation, and their ability to increase the reserved number. Thus, future studies with semi-field and field releases can be conducted to better understand the adaptation abilities of these parasitoids. 

## Figures and Tables

**Figure 1 insects-12-00924-f001:**
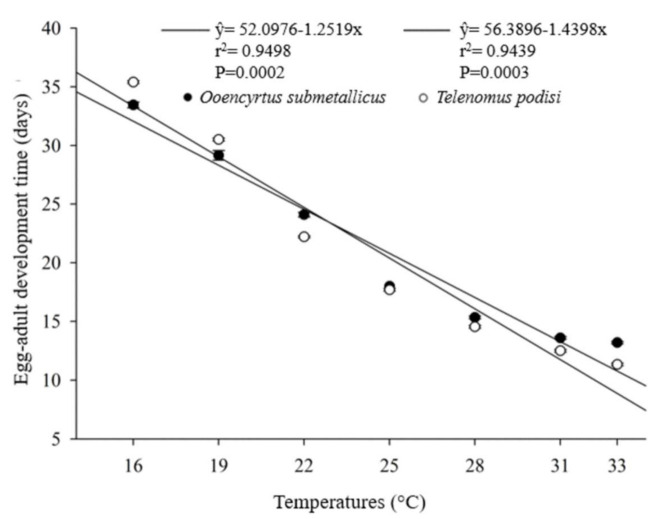
Egg–adult developmental time (days) of *Ooencyrtus submetallicus* and *Telenomus podisi* in *Euschistus heros* eggs at different temperatures.

**Figure 2 insects-12-00924-f002:**
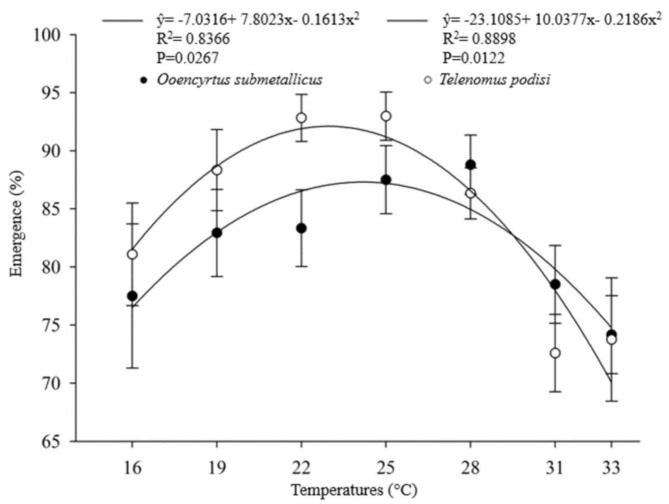
Emergence (%) (mean ± EP) of *Ooencyrtus submetallicus* and *Telenomus podisi* in *Euschistus heros* eggs at different temperatures.

**Figure 3 insects-12-00924-f003:**
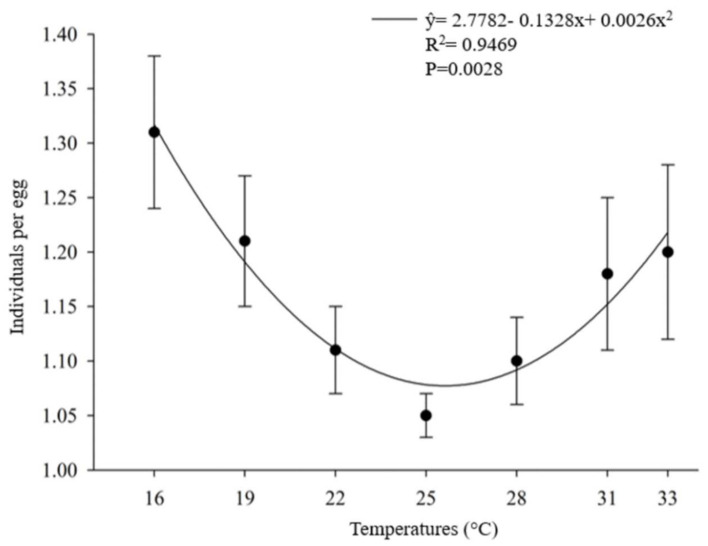
Number of parasitoids emerged per host egg parasitized (%) (mean ± EP) of *Telenomus podisi* in *Euschistus heros* eggs at different temperatures.

**Figure 4 insects-12-00924-f004:**
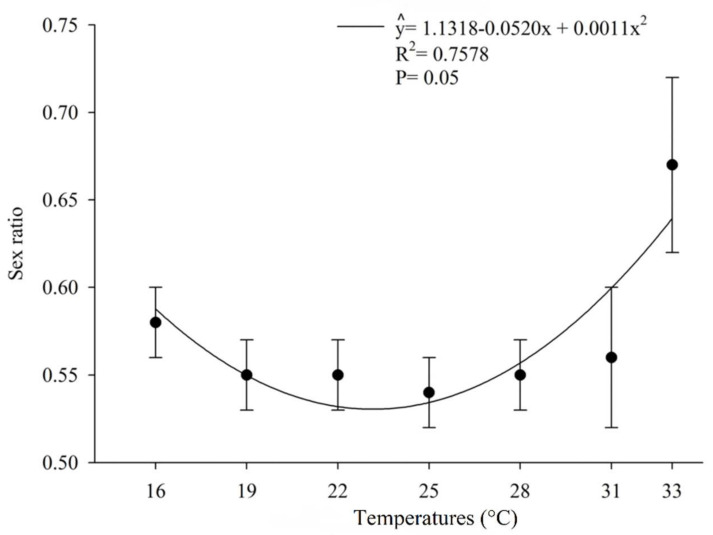
Sex ratio (%) (mean ± EP) of *Telenomus podisi* in *Euschistus heros* eggs at different temperatures.

**Figure 5 insects-12-00924-f005:**
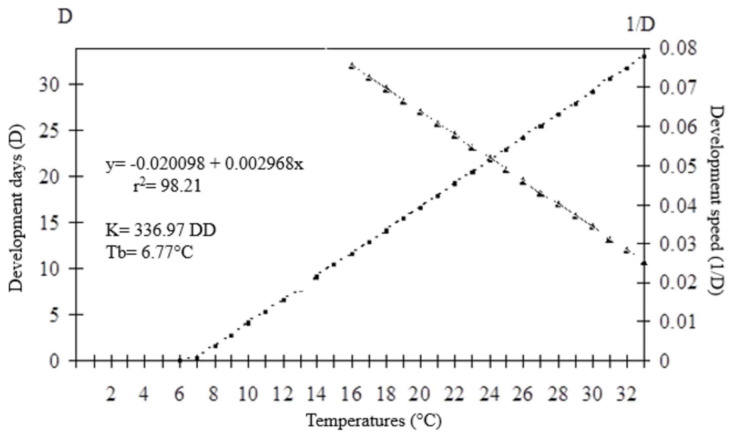
Duration (days) and developmental speed of *Ooencyrtus submetallicus* in eggs *of Euschistus heros* at different temperatures. K: thermal constant; Tb: base temperature.

**Figure 6 insects-12-00924-f006:**
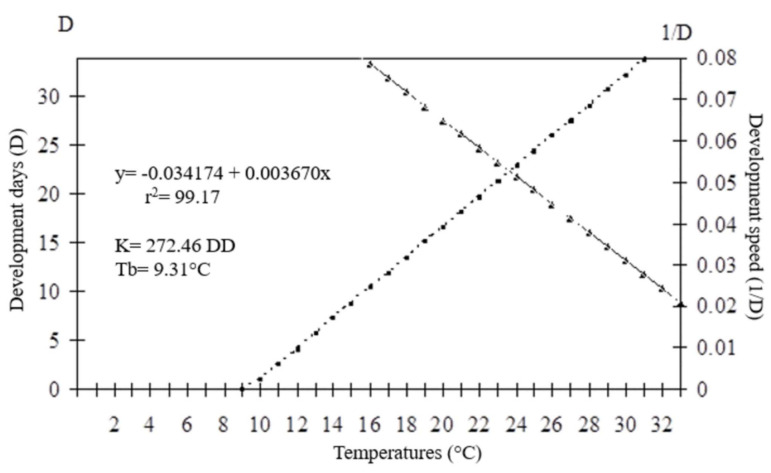
Duration (days) and developmental speed of *Telenomus podisi* in eggs *of Euschistus heros* at different temperatures. K: thermal constant; Tb: base temperature.

**Table 1 insects-12-00924-t001:** Biological characteristics (mean ± SE) of *Ooencyrtus submetallicus* and *Telenomus podisi* reared on eggs of *Euschistus heros.*

Characteristics	*O. submetallicus*	*T. podisi*	CV (%)
Cycle length (egg–adult) (days)	17.7 ± 0.10 a	17. 5 ± 0.11 a	2.72
Parasitism (%)	59.0 ± 1.43 a	62.5 ± 1.22 a	9.83
Emergence (%)	86.4 ± 2.26 a	87.3 ± 2.08 a	11.43
Sex ratio	1.0 ± 0.00 a	0.5 ± 0.02 b	6.76
Individuals emerged per egg (n)	1.9 ± 0.05 a	1.2± 0.04 b	12.11
Longevity without food (days)	9.6 ± 0.47 a	10.8 ± 0.51 a	21.57
Longevity with food (days)	18.1 ± 0.82 a	19.9 ± 0.83 a	19.60

Means followed by the same letter within a row do not differ by Student’s *t*-test at 5% probability.

**Table 2 insects-12-00924-t002:** Estimated number of generations per year of *Telenomus podisi, Ooencyrtus submetallicus,* and *Euschistus*
*hero*, considering climatic data for seven soybean-producing areas of Brazil.

Localities	*T. podisi*	*O. submetallicus*	*E. heros*
Tupanciretã, RS	11.43	11.99	4.05
Mal. Cândido Rondon, PR	13.12	13.36	5.46
Maracaju, MS	15.47	15.26	7.41
Dourados, MS	18.29	18.33	9.76
Ponta Porã, MS	15.29	16.39	8.57
Sorriso, MT	18.82	18.78	11.03
Rio Verde, GO	15.05	14.92	7.06

## Data Availability

The data presented in this study are openly available https://drive.google.com/drive/folders/1fv5D3LqWU5ljGMPKpoLU8QFNxScBuj-9?usp=sharing (accessed on 17 July 2021).

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
