# Peer review of "Thermal Requirements of Ooencyrtus submetallicus (Hym.: Encyrtidae) and Telenomus podisi (Hym.: Platygastridae) Parasitizing Euschistus heros Eggs (Hem.: Pentatomidae)"

_insects, 2021, doi:10.3390/insects12100924_

Round 1

Reviewer 1 Report

The authors have done a stellar job addressing all of my previous comments and those of other reviewers. I can recommend publication in Insects following some minor revisions I have had in mind below:

On lines 476 through 479 in the revised paper, the authors are emphasizing the need for future studies with fluctuating temperatures regimes to understand the real effect of temperature on life characteristics of parasitoids. This has been done previously and I'd suggest adding a references by J. Econ. Entomol. 112: 1560-1574 (ectoparasites BCAs reared for field applications at temperatures that fluctuate over 24-h cycles) and J. Econ. Entomol. 112:1062-1072 (endoparasites BCAs reared for field releases across a range of realistic temperatures experienced in the field) to support your statements. 

Also the references in the reference list are still not in the format of the journal and should be reformatted to match the style of the journal.

Good luck!    

Author Response

Dear Reviewer 1

Response: Suggestion accepted

1. Original sentence: “...However, it is important to emphasize that the results are obtained in the laboratory under constant temperatures, therefore, studies with fluctuating temperatures are necessary to understand the real effect of temperature on the characteristics of parasitoids, since this is the closest to the daily fluctuations of temperature that occur in the field.” End of sentence

1.1 Corrected sentence: “…studies with fluctuating temperatures are necessary to understand the real effect of temperature on the characteristics of parasitoids, since this is the closest to the daily fluctuations of temperature that occur in the field. As well as studies conducted for Diaphorencyrus aligarhensis (Shafee, Alam and Argarwall, 1975) (Hymenoptera: Encyrtidae) and Tamarixia radiata (Waterston, 1922) (Hymenoptera: Eulophidae), where fluctuating temperatures demonstrated significant effects on the life history of parasitoids, helping in the optimization of mass creations and releases, in addition to improving climate modeling predictions [39-40].

  1. References cited in the reference list have been reformatted to match the journal style.
  2. Sincerely, The authors.

Reviewer 2 Report

Dear,

Although the manuscript has undergone extensive corrections, some concerns need to be addressed before final acceptance.

Some of the suggestions based on the previous review process were not answered correctly or were not added in the text;

Lines 30 and 31: The estimated mean number of generations per year of the parasitoids ... The finding is for which temperature? All temperatures or some of them?

Response: The estimated average number of generations per year of the parasitoids is the mean temperatures of the last ten years in each region studied.

OK. Write it in the manuscript. 

Line 41: The sex ratio and the number of adult parasitoids that emerged per parasitized egg was superior for O. submetallicus. Is this result for all regions?

OK. Add "in all regions" in the sentence. 

Line 49: The estimated average number of generations per year for both parasitoids was greater than the pest E. heros. The finding is for which temperature?

Response: The analysis was conducted based on the last ten years of each studied region. With this, it was possible to estimate the average number of generations of each parasitoid and its host.

Please clearly write that mentioned findings are for which temperature?

Line 81 and 82: more details about the lowest and upper-temperature threshold, thermal constant, and optimum temperature are needed. Add adequate and brief definitions for them.

Response: Text changed to "Lower thermal threshold (base temperature), below which no development occurs; upper thermal threshold, which is the maximum growth temperature; optimal temperature, which generally corresponds to the fastest development and the greatest number of descendants; constant thermal (degree-days), which is the amount of accumulated degree-days necessary for the insect to pass from one stage to the other and, based on the temperature of the location under study, estimate the number of generations that can be obtained under environmental certain conditions [...]”

A good explanation but please note that the sentence is too long. Rewrite it in two or three separate sentences. 

Line 125: To what ratio were the foods mixed?

Response: Generally, the food was provided in a Petri dish (100×20 mm) according to the number of stinkbugs present in the plastic pots that they were raised and the exact proportion was not measured.

The ratio of utilized foods should be written. 

Please also consider the following new comments for the revised version of the manuscript:

In Fig. 1 and 2, write the p and r2 values in the correct format. For example, p = 0.0002 not 0,0002.

The scientific names throughout the manuscript should be italicized even in the title and sub-titles. 

The conclusion section should be strengthened. Briefly state the objectives of the research, explain the results obtained and their application, and add suggestions for future research.

Best regards

Author Response

Dear Reviewer 2

I appreciate your valuable comments and suggestions for improving the quality of our manuscript. After analyzing the suggestions and questions, as well as the errors indicated and the comments contained, we made the changes as indicated in coverletter.

Thank you in advance for your collaboration.

Sincerely, The authors.

This manuscript is a resubmission of an earlier submission. The following is a list of the peer review reports and author responses from that submission.

Round 1

Reviewer 1 Report

These are my main comments on the MS (insects-1348411) entitled: " Thermal Requirements of Two Egg Parasitoids of Brown Stinkbug Euschistus heros (Hem.: Pentatomidae) " by Valeria Freitas Chaves and colleagues.

The authors have graphed and presented their results clearly, drawing some attention to the implications of their findings. I found the study of interest and a good contribution to the knowledge of bioecology of insect parasitoids. The methods used are appropriate for the objectives of the work and, in general, well depicted. The resulting figures are sufficient, informative, and of good quality helping to follow the reasoning throughout the manuscript.

Major comments:

The Intro and Discussion provide no insight on how this MS relates to the various other ones cited in the text or concerns that have been raised by other researchers. The authors do not present any hypotheses or expectations that could be connected to previous studies; adding these details will improve the paper. This article should provide details on all these fronts to provide the proper context for the work.

My major concern is that the authors are extrapolating the applicability of their results beyond what the design supports. These are only data from a set of highly artificial laboratory conditions at constant temperatures, so the inference power of the paper is very limited, but authors do not acknowledge this detail at all and need to be more forthcoming. This is a critical limitation of the study, and the authors must concede and discuss this. It creates major bias in your model fitting and validation analysis. It is well known that most of laboratory experiments are conducted under constant temperatures whereas in nature daily temperature fluctuations can be very wide. The interaction of cyclic temperatures with nonlinear development or life history parameters can introduce significant deviations from the parameters developed here, especially at the lower and higher temperatures of the development rate, viability, and reproductive activity functions. The consequences of getting this wrong will affect real people and livelihoods. So, I am suggesting to the authors to tone-down the language a little and admit that there are still substantive uncertainties to be considered, including uncertainty as to how generalizable the results are to open field conditions.

The study is also divorced from other similar studies on rearing BCAs in microcosms for field application. Some of the authors statement would be much stronger if they tie their work to the body of literature that has built up from rearing parasitoid agents in microcosms at temperature regimes that fluctuate over 24h cycles (see examples from McCalla et al. 2019 J. Econ. Entomol. 112: 1560-1574; Milosavljevic et al. 2019 J. Econ. Entomol. 112:1062-1072). I’d suggest adding these references to your discussion. This is not to diminish the data gathered in this study, they are of value. But it is important for the authors not to overgeneralize, and to warn the reader, including regulatory agencies, against doing so as well. Adding these details will improve the paper.

Overall, I was excited to see the results of the paper after reading the abstract, but I found it hard to extract key messages useful to policymakers and professionals, probably in large part due to the lack of connection with other published work and need for improved structure of the current manuscript.

The next draft of this paper will need to be dramatically different to have a chance at publication in my opinion.

Author Response

Dear reviewer

I appreciate your valuable comments and suggestions for improving the quality of our manuscript. After analyzing the suggestions and questions, as well as the errors indicated and the comments contained, we made the changes as indicated below.

We agree with the reviewer that there are still uncertainties about the results obtained, especially because they are laboratory studies under constant temperatures.  Therefore, studies with fluctuating temperatures are still necessary to understand the real effect of temperature on the characteristics of the parasitoids, as this is the closest to the daily temperature fluctuations that occur in the field.

The English in this article has been proofread and the grammar revised by a native English
Speaker. 

Thank you in advance for your collaboration.

Fabricio Fagundes Pereira

Universidade Federal da Grande Dourados.

Reviewer 2 Report

Dear,

The manuscript with the title of ‘Thermal Requirements of Two Egg parasitoids of Brown Stink bug Euschistus heros (Hem.: Pentatomidae)’ has an interesting subject and applicable findings. The experiments have the correct design and the results were properly analyzed. However, vague sentences and incorrect formatting problems can be found throughout the manuscript, which should be modified before publication.    

The title should include the main contents of the manuscript. In the initial parts of the research, the effect of temperature was not considered. The title needs to be rewritten.

Line 24: ‘Our objectives were’.

Line 26: use ‘O.’ instead of ‘Ooencyrtus’.  Use abbreviated names after the first mention of all scientific names.

Line 26: Ooencyrtus submetallicus produced more females and individuals per egg compared to T. podisi. Was the mentioned production documented in which area? The sentence is vague.

Lines 29 and line 30: define the temperature thresholds (Tb) and the thermal constants (K) in the first mention.

Lines 30 and 31: The estimated mean number of generations per year of the parasitoids ... The finding is for which temperature? All temperatures or some of them?

Line 41: The sex ratio and the number of adult parasitoids that emerged per parasitized egg were superior for O. submetallicus. Is this result for all regions?

Line 49: The estimated average number of generations per year for both parasitoids was greater than the pest E. heros. The finding is for which temperature?

Line 70: and E. heros.

Line 73: add more abiotic factors and emphasize temperature.

Line 81 and 82: more details about the lowest and upper-temperature threshold, thermal constant, and optimum temperature are needed. Add adequate and brief definitions for them.

Line 89: remove ‘On the other hand,’.

Line 96: Is it possible to write the test place in English?

Lines 101, 113, and 121: Italicize the scientific names. Check all names throughout the manuscript.

Lines 108 and 109: Rewrite the sentence in high clarity.

Line 105: What is the BOD? Explain it.

Lines 111 and 115: It is not clear why the name of the researcher or author is written. Rewrite this sentence and delete the researcher and author names. You can thank your colleague in the acknowledgment section.

Lines 113–120: It would be better to write the steps of breeding and identification of parasitoids in one paragraph.

Line 125: To what ratio were the foods mixed?

Lines 179-181: Rewrite the sentence. Do these question marks have a special meaning?

Table 1: The order and family names and rearing conditions are not needed. All are repeated.

Line 206: Remove the dot from the end of the sub-title.

Fig 1: ‘Egg-to-adult developmental time’ is better than ‘duration of the cycle’ in the Y-axis. Further, the scientific names are duplicated, and the ‘Observed’ word is not needed. The order and family names in the figure title should be removed. Consider this comment for all figures.

Fig. 2. Consider the previous comment for this figure as well. Write emergence and temperature in English.

Line 211: More explain the emergence of both parasitoids. The emergence of both parasitoids was increased from 16° to 25 ° C and then decreased.

Line 225: extreme temperatures! Use more appropriate words and rewrite the sentence.

Figure 5: Explain K and Tb in the figure title.

Table 2 is fully incorrect. E. heros was written in the egg parasitoid column.

The discussion is good but more explanation about the application of present findings is necessary. Make suggestions for the possibility of using the method and the actual results in future researches.

The references are not in the format of the journal. All must be reformatted.

All the bests

Author Response

Dear reviewer

I appreciate your valuable comments and suggestions for improving the quality of our manuscript. After analyzing the suggestions and questions, as well as the errors indicated and the comments contained, we made the changes as indicated below.

Thank you in advance for your collaboration.

Fabricio Fagundes Pereira

Universidade Federal da Grande Dourados.

The manuscript with the title of ‘Thermal Requirements of Two Egg parasitoids of Brown Stink bug Euschistus heros (Hem.: Pentatomidae)’ has an interesting subject and applicable findings. The experiments have the correct design and the results were properly analyzed. However, vague sentences and incorrect formatting problems can be found throughout the manuscript, which should be modified before publication.    

The title should include the main contents of the manuscript. In the initial parts of the research, the effect of temperature was not considered. The title needs to be rewritten.

Line 24: ‘Our objectives were’.

Line 26: use ‘O.’ instead of ‘Ooencyrtus’.  Use abbreviated names after the first mention of all scientific names.

Line 26: Ooencyrtus submetallicus produced more females and individuals per egg compared to T. podisi. Was the mentioned production documented in which area? The sentence is vague.

Lines 29 and line 30: define the temperature thresholds (Tb) and the thermal constants (K) in the first mention.

Lines 30 and 31: The estimated mean number of generations per year of the parasitoids ... The finding is for which temperature? All temperatures or some of them?

Line 41: The sex ratio and the number of adult parasitoids that emerged per parasitized egg were superior for O. submetallicus. Is this result for all regions?

Line 49: The estimated average number of generations per year for both parasitoids was greater than the pest E. heros. The finding is for which temperature?

Line 70: and E. heros.

Line 73: add more abiotic factors and emphasize temperature.

Line 81 and 82: more details about the lowest and upper-temperature threshold, thermal constant, and optimum temperature are needed. Add adequate and brief definitions for them.

Line 89: remove ‘On the other hand,’.

Line 96: Is it possible to write the test place in English?

Lines 101, 113, and 121: Italicize the scientific names. Check all names throughout the manuscript.

Lines 108 and 109: Rewrite the sentence in high clarity.

Line 105: What is the BOD? Explain it.

Lines 111 and 115: It is not clear why the name of the researcher or author is written. Rewrite this sentence and delete the researcher and author names. You can thank your colleague in the acknowledgment section.

Lines 113–120: It would be better to write the steps of breeding and identification of parasitoids in one paragraph.

Line 125: To what ratio were the foods mixed?

Lines 179-181: Rewrite the sentence. Do these question marks have a special meaning?

Table 1: The order and family names and rearing conditions are not needed. All are repeated.

Line 206: Remove the dot from the end of the sub-title.

Fig 1: ‘Egg-to-adult developmental time’ is better than ‘duration of the cycle’ in the Y-axis. Further, the scientific names are duplicated, and the ‘Observed’ word is not needed. The order and family names in the figure title should be removed. Consider this comment for all figures.

Fig. 2. Consider the previous comment for this figure as well. Write emergence and temperature in English.

Line 211: More explain the emergence of both parasitoids. The emergence of both parasitoids was increased from 16° to 25 ° C and then decreased.

Line 225: extreme temperatures! Use more appropriate words and rewrite the sentence.

Figure 5: Explain K and Tb in the figure title.

Table 2 is fully incorrect. E. heros was written in the egg parasitoid column.

The discussion is good but more explanation about the application of present findings is necessary. Make suggestions for the possibility of using the method and the actual results in future researches.

The references are not in the format of the journal. All must be reformatted.

Response: The title was changed to "Thermal Requirements of Ooencyrtus submetallicus (Hym.: Encyrtidae) and Telenomus podisi (Hym.: Platygastridae) Parasitizing Eggs of Euschistus heros (Hem.: Pentatomidae)"

Previous title: Thermal Requirements of Two Egg Parasitoids of Brown Stinkbug Euschistus heros (Hem.: Pentatomidae)

  1. Line 24.

Response: Suggestion accepted

Resposta: Sugestão aceita

  1. Line 26.

Response: Suggestion accepted

  1. Line 26:

Response: The comparative biology study was conducted in the laboratory, just to compare the characteristics of both parasitoids, and was not tested in the mentioned regions. However, the thermal requirements experiment was carried out based on the areas mentioned in the work. Thus, the sex ratio and the number of individuals per egg were higher for O. submetallicus at 25 °C in the laboratory.

  1. Lines 29 and line 30.

Response: We agree with the suggestion, but the temperature limit (Tb) and thermal constants (K) were defined throughout the introduction, so that the abstract is not too long.

  1. Lines 30 and 31.

Response: The estimated average number of generations per year of the parasitoids is the mean temperatures of the last ten years in each region studied.

  1. Line 41. Response: Yes
  2. Line 49. Response: The analysis was conducted based on the last ten years of each studied region. With this, it was possible to estimate the average number of generations of each parasitoid and its host.
  3. Line 70. Response: Suggestion accepted
  4. Line 73. Response: We changed sentence to "The efficiency, development and reproduction of these parasitoids can be affected by abiotic factors such as light, humidity and especially temperature, which influences the biological aspects the most [...]"
  5. Line 81 and 82:

Response: Text changed to "Lower thermal threshold (base temperature), below which no development occurs; upper thermal threshold, which is the maximum growth temperature; optimal temperature, which generally corresponds to the fastest development and the greatest number of descendants; constant thermal (degree-days), which is the amount of accumulated degree-days necessary for the insect to pass from one stage to the other and, based on the temperature of the location under study, estimate the number of generations that can be obtained under environmental certain conditions [...]”

  1. Line 89. Response: Suggestion accepted
  2. Line 96. Response: Yes

 “The experiments were carried out in the premises of the Laboratory for Biological Control of Insects (LECOBIOL) (22°19’80’’S and 54°93’38’’W), belonging to the Faculty of Biological and Environmental Sciences (FCBA) of the Federal University of Grande Dourados (UFGD), in Dourados, Mato Grosso do Sul, Brazil.”

  1. Lines 101, 113 and 121:

Response: Suggestion accepted

  1. Lines 108 e 109.

Response: Phrase was changed to “The species of this parasitoid was found in E. meditabunda eggs on leaves of tomato plants, Solanum lycopersicum (Linnaeus, 1753) (Solanales: Solanaceae). The collection was conducted by Antonio de Souza Silva in the region of Dourados, MS.”

Original sentence: “Individuals were initially collected parasitizing eggs of E. meditabunda from tomato plants, Solanum lycopersicum (Linnaeus, 1753) (Solanales: Solanaceae) located in Dourados, MS (INSERE COORDENADAS). The collection was made by Antônio de Souza Silva, and specimens were identified by one of the authors (V.A.C.) according to Noyes (2010).”

Frase anterior: “ Indivíduos foram inicialmente coletados parasitando ovos de E. meditabunda de tomateiro, Solanum lycopersicum (Linnaeus, 1753) (Solanales: Solanaceae), localizado em Dourados, MS. A coleta foi realizada por Antonio de Souza Silva e os espécimes foram identificados por um dos autores (V.A.C) segundo Noyes (2010).”

  1. Line 105:

Response: B.O.D (Biochemical Oxygen Demand) Greenhouse Incubator or B.O.D chamber are laboratory equipment, developed for work that requires exact temperature control.

  1. Lines 111 and 115: Response: Rewritten sentence "The collection was conducted by Antonio de Souza Silva in the region of Dourados, MS."

Previous sentence “The collection was made by Antônio de Souza Silva, and specimens were identified by one of the authors (V.A.C.) according to Noyes (2010).”

  1. Lines 113-120: Response: Suggestion accepted
  2. Lines 125. Response: Generally, the food was provided in a Petri dish (100×20 mm) according to the number of stinkbugs present in the plastic pots that they were raised and the exact proportion was not measured.
  3. Lines 179-181: Response: Suggestion accepted. The question marks were typos.
  4. Table 1:

Response: Suggestion accepted

  1. Line 206

Response: Suggestion accepted

  1. Fig 1. Response: Tables and figures were changed according to this comment.
  2. Fig. 2. Response: Change made
  3. Line. 211

Response: According to other studies, the exposure of parasitoids to lower and higher temperatures throughout their development negatively affects their life history, which may be a consequence of the proximity of the lethal temperature for their development, thus affecting their emergence. The temperature of 25 °C can be considered optimal, where the greatest development occurs and the greatest number of offspring emerged (Torres et al., 1997; Krugner et al., 2007; Pastori et al., 2011).

  1. Line 225: Response: Rewritten sentence: “The number of individual O. submetallicus per host egg was not influenced by the temperatures tested. On the other hand, the most individual T. podisi occurred at 16 °C, with an average of 1.31 parasitoids emerging per parasitized egg (Figure 3).”

Previous sentence “The number of offspring produced per parasitized host egg by O. submetallicus was not influenced by the temperature regimes with an average of 1,31 parasitoids/egg.  On the other hand, the temperature regime studied affected the number of individual T. podisi per host egg parasitized (Figure 3).

  1. Figure 5: Response: Thermal constant and base temperature were defined in the figure titles
  2. Table 2

Response: Again, it was a typo

  1. Response: In the discussion, we agree that more work was needed to discuss the real effects of temperature, as well as the uncertainties that still exist. Thus, changes were made according to the suggestions and can be observed throughout the manuscript.

Response: The article was  reformatted according to the journal's guidelines.

We await the final answer and are available to clarify any questions.

Sincerely,

The authors.
